# Topological Data Analysis in Cardiovascular Signals: An Overview

**DOI:** 10.3390/e26010067

**Published:** 2024-01-12

**Authors:** Enrique Hernández-Lemus, Pedro Miramontes, Mireya Martínez-García

**Affiliations:** 1Computational Genomics Division, National Institute of Genomic Medicine, Mexico City 14610, Mexico; pmv@ciencias.unam.mx; 2Center for Complexity Sciences, Universidad Nacional Autónoma de México, Mexico City 04510, Mexico; 3Department of Mathematics, Sciences School, Universidad Nacional Autónoma de México, Mexico City 04510, Mexico; 4Department of Immunology, National Institute of Cardiology, Mexico City 14080, Mexico; mireya.martinez@cardiologia.org.mx

**Keywords:** topological data analysis, cardiovascular signals, alegbraic topology, persistent homology, mapper algorithm

## Abstract

Topological data analysis (TDA) is a recent approach for analyzing and interpreting complex data sets based on ideas a branch of mathematics called algebraic topology. TDA has proven useful to disentangle non-trivial data structures in a broad range of data analytics problems including the study of cardiovascular signals. Here, we aim to provide an overview of the application of TDA to cardiovascular signals and its potential to enhance the understanding of cardiovascular diseases and their treatment in the form of a literature or narrative review. We first introduce the concept of TDA and its key techniques, including persistent homology, Mapper, and multidimensional scaling. We then discuss the use of TDA in analyzing various cardiovascular signals, including electrocardiography, photoplethysmography, and arterial stiffness. We also discuss the potential of TDA to improve the diagnosis and prognosis of cardiovascular diseases, as well as its limitations and challenges. Finally, we outline future directions for the use of TDA in cardiovascular signal analysis and its potential impact on clinical practice. Overall, TDA shows great promise as a powerful tool for the analysis of complex cardiovascular signals and may offer significant insights into the understanding and management of cardiovascular diseases.

## 1. Introduction: Data Analytics in Modern Cardiology

Cardiovascular diseases (CVDs) have been, for a long time, a major global health problem that has caused more deaths than any form of cancer or respiratory disease combined. The detection and prediction of CVDs is made difficult by the numerous etiological factors, complex disease pathways, and diverse clinical presentations [1,2]. However, with the advent of an enhanced capability for the generation of complex high-dimensional data from electronic medical records, mobile health devices, and imaging data, one is presented with both challenges and opportunities for data-driven discovery and research. While traditional statistical approaches for risk stratification have been broadly developed, leading to important improvements of diagnosis, prognosis and in some cases therapeutics, most of these models have limitations in terms of individualized risk prediction. Recently, data analytics, artificial intelligence and machine learning have emerged as potential solutions for overcoming the limitations of traditional approaches in the field of CVD research.

Advance analytics algorithms can have a major impact on cardiovascular disease prediction and diagnosis. CVD data, however, remain challenging for common machine learning and data analytics approaches due to the wide variety and large heterogeneity of the diverse cardiovascular signals currently being probed [3,4]. Among the several different approaches that are arising to cope with such disparate data, one that results particularly outstanding for its generality and its ability to handle data integrating diverse dynamic ranges and scales is topological data analysis [5,6].

Topological data analysis (TDA) is, in short, a family of analytic methods that has been gaining relevance and recognition to model, predict and understand the behavior of complex biomedical data. TDA is founded on the tenets of algebraic topology, a mathematical field that deals with the *shape* of data and has a set of methods for studying it [7]. In this review article, we want to present the fundamentals of TDA and its applications in the analysis of cardiovascular signals. We aim to make these techniques accessible to non-experts by explaining their theoretical foundations and surveying their use in computational cardiology. We also discuss the limitations of these methods and suggest possible ways to incorporate them into clinical care and biomedical informatics in the context of cardiovascular diseases.

Figure 1 presents a graphic overview of the main ideas, starting with one or several data sources on cardiovascular signals coming from medical devices, wearables, clinical monitors, electronic health records and so on. Data are used to generate data clouds that are turned into *Metric data sets* that are then processed and analyzed with the tools of topological data analysis (see below) to generate homology groups, persistence diagrams and signatures useful to classify signals for a deeper understanding of their phenomenology.

## 2. Fundamentals of Topological Data Analysis

Topology is a branch of mathematics that deals with the shapes of objects. It provides a framework for understanding how objects can be deformed and still retain their essential properties. For example, a circular rubber band can be stretched into an oval, but a segment of string cannot. Topologists study the connectedness of objects by counting their number of pieces and holes, and use this information to classify objects into different categories.

A related field, algebraic topology, provides a framework for describing the global structure of a space in a precise and quantitative way. It uses methods that take into account the entire space and its objects rather than just local information. Algebraic topology uses the concept of homology to classify objects based on their number and type of holes, and topological spaces, which consist of points and neighborhoods that satisfy certain conditions to do so. The notion of a topological space allows for flexibility in using topological tools in various applications, as it does not rely on numerical values to determine the proximity of points, but rather whether their neighborhoods overlap.

More formally, an *Homology* is a set of topological invariants represented by homology groups $H_{k}\left(X\right)$ that describe *k*-dimensional holes in topological space *X*. The rank of $H_{k}\left(X\right)$ (known as the *k*th *Betti number*), is analogous to the dimension of a vector space and indicates the number of *k*-dimensional holes. For example, $H_{0}$ corresponds to zero-dimensional features or connected components, $H_{1}$ corresponds to one-dimensional features or cycles, and $H_{2}$ corresponds to two-dimensional features or cavities. It is also possible to study $H_{k}$ for larger values of *k*, but it becomes more difficult to *visualize* the associated features.

At this stage, homology seems to be a rather abstract concept; however, it can be connected in a straightforward manner to data analytics once we recognize one quite important property of data points: their *shape* as a set, that is, the way in which data points are distributed in the feature’s space. One can obtain an approximation to this shape by looking at how *holes* are distributed in the data space. Our understanding of why points accumulate in one region of the data space and are missing in other regions is a powerful tool to look for trends in the data. In order to understand the topological shape of a data set and identify its holes, it is useful to assign a topological structure to the data and calculate topological invariants. Homology groups are useful for this purpose because there are efficient algorithms for computing some of the more relevant of these invariants in the context of TDA [8].

Homology groups are hence used to classify topological spaces based on the topological features of their shape, such as connectedness, loops, and voids. Homology groups of a topological space are *invariant under continuous deformations*, meaning that if two spaces have different homology groups, then they *cannot be continuously deformed into one another and are therefore topologically distinct*. Homology can thus be used to distinguish between spaces that may *appear* to be the same from other perspectives, such as those that have the same dimension or the same symmetries.

In the context of topological data analysis (TDA), the interpretation of topological features like connectedness, loops, holes, and voids involves understanding the geometric and structural properties of the data that these features represent. Let us briefly review some of these ideas.

**Connectedness**: Connectedness refers to the property of data points or regions being connected in a topological space. In TDA, connectedness typically corresponds to the number of connected components in a data set. The number of connected components can provide insights into the overall structure of the data. High connectedness implies that the data are relatively well-connected, while low connectedness may indicate separate clusters or isolated data points.

**Loops**: Loops represent closed paths or cycles in the data. They can occur when points or regions in the data form closed curves or circles. Loops can capture repetitive or periodic patterns in the data. They are often associated with cyclic structures or data points arranged in circular or ring-like formations.

**Holes**: Holes correspond to empty spaces or voids in the data where there are no data points. These voids can take various shapes, including spherical voids, tunnel-like voids, or irregular voids. The presence and characteristics of holes provide information about data *emptiness*. They can indicate the absence of data in specific regions or reveal patterns in data distribution, such as clustering around voids.

**Voids**: Voids are regions of space that lack data points. They are similar to holes but can be more generalized and may not necessarily be enclosed by data. Voids are often used to study the spatial distribution and density of data points. Large, persistent voids may suggest regions where data are scarce, while small, transient voids may highlight local fluctuations.

To interpret these topological features effectively, TDA often employs persistence diagrams or barcode diagrams. These diagrams summarize the *births* and *deaths* of topological features across a range of spatial scales, providing a quantitative way to assess the significance and persistence of these features. Here is how persistence diagrams relate to the interpretation of topological features:

**Connectedness**: The number of connected components is quantified by points in the persistence diagram. Longer persistence (vertical distance from birth to death) indicates more robust connected components.

**Loops**: Loops are associated with features in the persistence diagram. Longer loops correspond to more persistent cyclic patterns in the data.

**Holes and Voids**: Holes and voids are represented by clusters of points in the persistence diagram. The position of points in the diagram indicates the spatial scale and persistence of these features.

In summary, interpreting topological features in TDA involves understanding the presence, size, and persistence of connectedness, loops, holes, and voids in user data. Persistence diagrams provide a concise visual representation of these features and their characteristics across different scales, aiding in the exploration and analysis of complex data sets. A more formal explanation of these concepts is discussed in Section 2.1.

### 2.1. Persistent Homology

We discuss some of the main homology groups used for data analytics. We start by presenting the *Persistent Homology Group* or Persistent Homology, PH.

PH identifies topological features of a space at different scales. Features that are consistently detected across a broad range of scales are considered more likely to be true features of the space rather than being influenced by factors such as sampling errors or noise. To use persistent homology, the space must be represented as a *simplicial complex* (i.e., a set of *polytopes*, like points, line segments, triangles, tetrahedra and so on) and a *filtration*, or a nested sequence of increasing subsets, must be defined using a distance function on the space.

To clarify such abstract concepts as simplicial complex and filtration, let us consider a set of measurements of some property (or properties) of interest in terms of the associated features for each point (see Figure 2A). We customarily call this set a *point cloud*; this represents the data. Point clouds, which, as we said, are simply collections of points, do not have many *interesting* topological properties per se. However, we can analyze their topology by placing a ball of radius ϵ around each point. This method (called filtration) allows for us to encode geometric information by increasing the value of ϵ, which determines how much the boundaries of the shape blur and expand. As ϵ increases (Figure 2B–E), points that were initially distinct may begin to overlap, altering the concept of proximity. Essentially, this process involves taking, let us say, *an impressionist* painting (when we partially close our eyes to reveal details, like we do to appreciate a picture from Claude Monet) looking at the point cloud to offer it a more defined shape.

A bit more formally, a simplicial complex is a collection of finite sets of points, called vertices, that are connected by edges, line segments connecting two vertices, and faces, which are polygons with three or more edges. The vertices, edges, and faces of a simplicial complex must satisfy certain conditions:Every face of the complex must be a simplex, that is, a triangle or a higher-dimensional analogue of a triangle.Every face of the complex must be a subset of one of the vertices of the complex.If a face of the complex is a subset of another face, then the larger face must be a subset of one of the vertices of the complex.

Once we learn to build a simplicial complex for a given scale (value of ϵ) by changing the value of ϵ, what we do is create a *filtered simplicial complex* (FSC). Every topological property (such as the homologies $H_{k}$) that persists through the FSC is a PH. Intuitively, different phenomena under study give rise to different point clouds that, when analyzed via an FSC, have different PHs.

#### 2.1.1. Building the FSC

By building the PH to a given point cloud, one aims to create a complex that approximates the original manifold using the given points. To achieve this, connections are established between points by adding edges between pairs of points; faces between triples of points; and so on. To determine which connections to create, we introduce a parameter called the filtration value (the ϵ we already mentioned), which limits the maximum length of the edges that can be included in our simplices. We vary ϵ and build the complex at each value, calculating the homology of the complex at each step.

There are three main strategies for using ϵ to assign simplices: the *Vietoris–Rips* strategy, the *witness* strategy, and the *lazy-witness* strategy [9]. The Vietoris–Rips strategy adds an edge between two points if their distance is less than ϵ, a face between three points if their pairwise distance is less than ϵ, and so on. This approach is accurate but computationally expensive. The witness strategy uses two sets of points, called *landmark points* and *witness points*, to create the complex. Landmark points are used as vertices and edges are added between two landmark points if there is a witness point within distance ϵ of both points, faces are added if there is a witness point within ϵ of all three points, and so on. The lazy-witness strategy is similar to the witness strategy in terms of how edges are assigned, but simplices of higher order are added anywhere there are *n* points that are all connected by edges.

#### 2.1.2. Calculating the PH

Once we choose a filtration, it is possible to calculate the homology groups (the $H_{k}$’s) of each space in the filtration. Homology groups are a way of describing the topological features of a space, such as connected components, holes, and voids. Depending on the particular task, we may choose a maximum value of *k* to build the first *k* homology groups. Then, we can use these homology groups to create a *barcode* or *persistence diagram*, which shows how the topological features of the space change as the scale changes [9].

To calculate persistent homology, it is possible to use a variety of algorithms, such as the already mentioned Vietoris–Rips complex, the Čech complex, or the alpha complex. These algorithms construct a simplicial complex from the data, which can then be used to calculate the homology groups of the space.

Calculating persistent homology and interpreting the results is a non-trivial task. Several issues need to be considered and decisions need to be taken in every step of the process. We can summarize the process as follows:**Simplicial Complex Construction**: Begin by constructing a simplicial complex from your data. This complex can be based on various covering maps, such as the Vietoris–Rips complex, yhe Čech complex, or the alpha complex, depending on the chosen strategy (see Section 2.4 and Section 2.5, as well as Table 1 below).The simplicial complex consists of vertices (0—simplices), edges (1—simplices), triangles (2—simplices), and higher-dimensional simplices. The choice of the complex depends on user data and the topological features of interest.**Filtration**: Introduce a filtration parameter (often denoted as ϵ) that varies over a range of values. This parameter controls which simplices are included in the complex based on some criterion (e.g., distance threshold).As ϵ increases, more simplices are added to the complex, and the complex evolves. The filtration process captures the topological changes as ϵ varies.**Boundary Matrix**: For each value of ϵ in the filtration, compute the boundary matrix (also called the boundary operator) of the simplicial complex. This matrix encodes the relations between simplices.Each row of the boundary matrix corresponds to a (*k* − 1)-dimensional simplex, and each column corresponds to a *k*-dimensional simplex. The entries indicate how many times a (k − 1)-dimensional simplex is a face of a *k*-dimensional simplex.**Persistent Homology Calculation**: Perform a sequence of matrix reductions (e.g., Gaussian elimination) to identify the *cycles* and *boundaries* in the boundary matrix.A cycle is a collection of simplices whose boundaries sum to zero, while a boundary is the boundary of another simplex.Persistent homology focuses on tracking the birth and death of cycles across different values of ϵ. These births and deaths are recorded in a persistence diagram or a barcode.**Persistence Diagram or Barcode**: The persistence diagram is a graphical representation of the births and deaths of topological features (connected components, loops, voids) as ϵ varies.Each point in the diagram represents a topological feature and is plotted at birth (*x*-coordinate) and death (*y*-coordinate).Interpretation:A point in the upper-left quadrant represents a long-lived feature that persists across a wide range of ϵ values.A point in the lower-right quadrant represents a short-lived feature that exists only for a narrow range of ϵ values.The diagonal represents features that are consistently present throughout the entire range of ϵ values.The distance between the birth and death of a point in the diagram quantifies the feature’s *persistence* or *lifetime*. Longer persistence indicates a more stable and significant feature.**Topological Summaries**: By examining the persistence diagram or a barcode, information can be extracted about the prominent topological features in a user data set.Features with longer persistence are considered more robust and significant.The number of connected components, loops, and voids can be quantified by counting points in specific regions of the diagram.

### 2.2. The Mapper Algorithm

Mapper is a recently developed algorithm that provides a reliable tool for topological data analysis. Mapper allows for researchers to identify and visualize the structure of a data set by creating a graph representation of the data [10].

The Mapper algorithm consists in the following steps [11] (see Figure 3):Covering the data set: The original data set (Figure 3a) is partitioned into a number of overlapping subsets called *nodes* (Figure 3b). This is accomplished using a function called the *covering map*. The covering map assigns each point in the data set to a node. Since the nodes are allowed to overlap, every point potentially belongs to multiple nodes.There are several different ways to define a covering map. The choice of a covering map, however, can significantly affect the resulting Mapper graph. Some common approaches to define a covering map include:(a)Filtering: The data set is partitioned based on the values of one or more variables. A data set may, for instance, be partitioned based on the values of a categorical variable, such as gender or race.(b)Projection: Data set partitioning is performed by calculating the distance between points in the data set and using it as a membership criteria. This can be achieved using a distance function such as Euclidean distance or cosine similarity.(c)Overlapping intervals: The data set is partitioned into overlapping intervals, such as bins or quantiles. This can be useful for data sets that are evenly distributed or those having a known underlying distribution.The choice of covering map depends on the characteristics of the data set and the research question being addressed. It is important to choose a covering map that is appropriate for the data set and that will yield meaningful results.Clustering the nodes: The nodes are then clustered using a clustering algorithm, such as *k*-means or single-linkage clustering. The resulting clusters (Figure 3c) represent the topological features of the data set, and the edges between the clusters represent the relationships between the features.

The resulting graph (Figure 3d), called a *Mapper graph*, can be used to identify patterns and relationships in the data set that may not be apparent from other forms of visualization [12].

### 2.3. Multidimensional Scaling

In the context of topological data analysis, multidimensional scaling (MDS) is a method to visualize the relationships between a set of complex objects as projected in a lower-dimensional space. MDS is a versatile technique used in various fields for analyzing and visualizing relationships between objects. It can be used both in classic data analysis approaches and in conjunction with TDA methods. MDS works by creating a map of the objects in which the distance between said objects reflects (to a certain point) the dissimilarity between them [13]. MDS is often used along other techniques, like clustering, to analyze patterns in data sets that have a large number of variables. Multidimensional scaling can help identify relationships between objects that are not immediately apparent; hence, it is useful to visually explore complex data sets [14]. There are several different algorithms for performing MDS, including classical MDS, nonmetric MDS, and metric MDS.

Classical MDS is the most common method (see Figure 4). In a nutshell, we start with a set of *n* points in a space of high dimension (*m*), then we introduce a measure of similarity (or dissimilarity), for instance, a distance (such as the Euclidean distance), then we have a square symmetric matrix with n×n pairwise distances, and MDS is attained by performing Principal Coordinate Analysis (i.e., eigenvalue decomposition) of such matrix. The result is a set of lower-dimensional coordinates for *n* points. Hence, classic MDS is based on the idea of preserving the pairwise distances between objects in the projected low-dimensional map. Classical MDS finds the map that best preserves the distances between objects using an optimization algorithm. The Nonmetric MDS method is similar to classical MDS, but it does not assume that the dissimilarities between objects are metric—i.e., represented by a continuous scale; instead, it preserves the rank order of the dissimilarities between objects, instead of the absolute values. Metric MDS, conversely, is a variant of classical MDS based on *stress minimization*, that is, by considering the difference between the distances in the low-dimensional map and the dissimilarities in the data set. This method is used when the dissimilarities between objects can be represented by a continuous scale.

In general, classical MDS is the most widely used method, but nonmetric MDS and metric MDS may be more appropriate in certain situations.

#### How to Determine the Scaling Approach?

The choice between classical MDS, nonmetric MDS, and metric MDS depends ultimately on the characteristics of the data and the specific research question. Some general guidelines are as follows:Classical MDS:-**When to Use:** Classical MDS is suitable when using metric (distance) data that accurately represent the pairwise dissimilarities between objects. In classical MDS, the goal is to find a configuration of points in a lower-dimensional space (usually 2D or 3D) that best approximates the given distance matrix.-**Pros:** It preserves the actual distances between data points in lower-dimensional representation.It provides a faithful representation when the input distances are accurate.It is well-suited for situations where the metric properties of the data are crucial.-**Cons:** It assumes that the input distances are accurate and may not work well with noisy or unreliable distance data.It may not capture the underlying structure of the data if the metric assumption is violated.Nonmetric MDS:-**When to Use:** Nonmetric MDS is appropriate when using ordinal or rank-order data, where the exact distances between data points are not known, but their relative dissimilarities or rankings are available. Nonmetric MDS finds a configuration that best preserves the order of dissimilarities.-**Pros:** It is more flexible than classical MDS and can be used with ordinal data.It can handle situations where the exact distances are uncertain or difficult to obtain.-**Cons:** It does not preserve the actual distances between data points, so the resulting configuration is only an ordinal representation.The choice of a monotonic transformation function to convert ordinal data into dissimilarity values can affect the results.Metric MDS:-When to Use: Metric MDS can be used when using data that are inherently non-metric, but it is believed that transforming it into a metric space could reveal meaningful patterns. Metric MDS aims to find a metric configuration that best approximates the non-metric dissimilarities.-**Pros:** It provides a way to convert non-metric data into a metric representation for visualization or analysis.It can help identify relationships in the data that may not be apparent in the original non-metric space.-**Cons:** The success of metric MDS depends on the choice of the transformation function to convert non-metric data into metric distances.It may not work well if the non-metric relationships in the data are too complex or cannot be adequately approximated by a metric space.

In summary, the choice between classical, nonmetric, and metric MDS depends on the nature of user data and the goals of analysis. If metric data are accurate and preservation of the actual distances is desired, classical MDS is appropriate. If data are ordinal or dissimilarity measures are uncertain, nonmetric MDS may be more suitable. Metric MDS can be considered when it is desired to convert non-metric data into a metric space for visualization or analysis, but it requires careful consideration of the transformation function.

### 2.4. Choosing the Covering Map

When choosing a covering map in TDA, there are several characteristics of the data sets which are relevant to consider; among these, we can mention the following:**Data Dimensionality:** The dimensionality of the data under consideration is crucial. Covering maps should be chosen to preserve the relevant topological information in the data. For high-dimensional data, dimension reduction techniques may be applied before selecting a covering map.**Noise and Outliers:** The presence of noise and outliers in the data can affect the choice of a covering map. Robust covering maps can help mitigate the influence of noise and outliers on the topological analysis.**Data Density:** The distribution of data points in the feature space matters. A covering map should be chosen to account for variations in data density, especially if there are regions of high density and regions with sparse data.**Topological Features of Interest:** It is important to consider the specific topological features one is interested in analyzing. Different covering maps may emphasize different aspects of data topology, such as connected components, loops, or voids. The election of a covering map should align with particular research objectives.**Computational Efficiency:** The computational complexity of calculating the covering map should also be taken into account. Some covering maps may be computationally expensive, which can be a limiting factor for large data sets.**Continuous vs. Discrete Data:** It should be determined whether the data under analysis are continuous or discrete. The choice of a covering map may differ based on the nature of the data.**Metric or Non-Metric Data:** Some covering maps are designed for metric spaces, where distances between data points are well defined, while others may work better for non-metric or qualitative data.**Geometric and Topological Considerations:** The geometric and topological characteristics of user data should be considered. Certain covering maps may be more suitable for capturing specific geometric or topological properties, such as persistence diagrams or Betti numbers.**Domain Knowledge:** Domain-specific knowledge should be incorporated into user choice of a covering map. Understanding the underlying structure of the data can guide the user in selecting an appropriate covering map.**Robustness and Stability:** The robustness and stability of the chosen covering map shpuld be assessed. TDA techniques should ideally produce consistent results under small perturbations of the data or variations in sampling.

In practice, there are various covering maps and TDA algorithms available, such as Vietoris–Rips complexes, Čech complexes, and alpha complexes. The choice of covering map should be guided by a combination of these factors, tailored to the specific characteristics and goals of user data analysis. It may also involve some experimentation to determine which covering map best captures the desired topological features.

Different types of covering maps are thus best suited for different kinds of data. Some of the main covering maps used in TDA are presented in Table 1.

Ultimately, the choice of covering map depends on the specific characteristics of user data, such as dimensionality, metric properties, and the topological features of interest. It is often beneficial to experiment with different covering maps and parameters to determine which one best captures the desired topological information for a particular data set. Additionally, combining multiple covering maps and TDA techniques can provide a more comprehensive understanding of complex data sets.

### 2.5. Different Strategies for Topological Feature Selection

The Vietoris–Rips strategy, the witness strategy, and the lazy-witness strategy are some of the best-known TDA methods to capture and analyze the topological features of data sets. Each of these strategies has its own advantages and disadvantages.

#### 2.5.1. Vietoris–Rips (VR) Strategy

The main advantage of the VR strategy is its simplicity, since the VR complex is relatively easy to understand and implement. It connects data points based on a fixed distance threshold, which is quite intuitive. Another advantage of the VR lies on its widespread use, for there is a significant body of literature and software implementations available. It works well with data in metric spaces where distances between points are well defined.

One of the disadvantages is that VR is quite sensitive to certain parameters; the choice of distance threshold (radius parameter) can significantly impact the topology of the resulting complex. Selecting an appropriate threshold can be challenging and may require prior knowledge of the data. VR can be also challenging for its computational burden: constructing the VR complex can be computationally expensive, especially for large data sets or high-dimensional data. VR is also limited in terms of robustness: the VR complex is sensitive to noise and outliers, and small perturbations in the data can lead to significant changes in complex topology.

#### 2.5.2. Witness Strategy (WS)

The witness strategy (WS), in turn, is more robust to noise and outliers compared to the VR complex. It selects a subset of *witness points* that can capture the topology of the data more effectively. WS is more flexible; witness complexes can be applied to both metric and non-metric data, making them versatile for various data types and able to handle data with varying sampling densities; this, they are suitable for irregularly sampled data sets.

Implementation of WS, however, can be more involved than that of the VR complex, as it requires selecting witness points and computing their witness neighborhoods. Also, while witness complexes are more robust, they still depend on parameters like the number of witness points and the witness radius. Choosing appropriate parameters can indeed be a non-trivial task.

#### 2.5.3. Lazy-Witness Strategy (LW)

The LW strategy is an optimization of the witness strategy that reduces computational cost. It constructs witness complex *on-the-fly* as needed, potentially saving memory and computation time. Like WS, the LW strategy is robust to noise and outliers.

In spite of these advantages, there are also shortcomings. Implementing the LW strategy can be more complex than implementing the basic witness strategy, as it requires careful management of data structures and computational resources. While it can be more memory efficient than precomputing a full witness complex, the LW strategy still consumes memory as it constructs the complex in real time. This may still be a limitation for very large data sets.

In summary, the choice between the Vietoris–Rips strategy, witness strategy, and lazy-witness strategy depends on the specific characteristics of user data and the computational resources available. The Vietoris–Rips complex is straightforward but sensitive to parameter choice and noise. The witness strategy offers improved robustness but may require more effort in parameter tuning. The lazy-witness strategy combines robustness with some memory and computation efficiency, making it a good choice for large data sets. Experimentation and a deep understanding of user data characteristics are essential when selecting the most appropriate strategy for user TDA analysis.

## 3. Applications of TDA to Analyze Cardiovascular Signals

### 3.1. General Features

Aljanobi and Lee [5] recently applied the Mapper algorithm to predict heart disease. They selected nine significant features in each of the two UCI heart disease data sets (Cleveland and Statlog). The authors then used a tri-dimensional SVD filter to improve the filtering process. As a result, they observed an accuracy of 99.32% in the Cleveland data set and 99.62% in the Statlog data set in predicting heart disease.

Though not precisely *signals* but rather contextual string corpora (i.e., *texts*), TDA has also been applied to the analysis of structured and unstructured text in EHRs and clinical notes. This is achieved by first converting textual data into a suitable format for analysis. This can involve techniques such as tokenization, stemming, and removing stop words. Afterwards, one needs to represent the text as numerical vectors using methods like Term Frequency-Inverse Document Frequency (TF-IDF) or word embeddings (Word2Vec, GloVe). Once data are vectorized and tokenized, TDA can be applied. Lopez and coworkers [15], for instance, used the Mapper algorithm to classify distinctive subsets of patients receiving optimal treatments post-acute myocardial infarction (AMI) in order to identify high-risk subgroups of patients for having a future adverse event (AE) such as death, heart failure hospitalization, or recurrent myocardial infarction. A retrospective analysis of 31 clinical variables from the EHR of 798 AMI subjects was conducted at a single center. The subjects were divided into high- and low-risk groups based on their probability of survival without AEs at 1 year. TDA identified six subgroups of patients. Four of these subgroups, totaling 597 subjects, had a probability of survival without AEs that was greater than a one-fold change, and were considered low risk. The other two subgroups, totaling 344 subjects, had a probability of survival without AEs that was less than a one-fold change, and were considered high risk. However, 143 subjects (18% of the total) were classified as intermediate risk because they belonged to both high- and low-risk subgroups. TDA was also able to significantly stratify AMI patients into three subgroups with distinctive rates of AEs up to 3 years after AMI. This approach to EHR-based risk stratification does not require additional patient interaction and is not dependent on prior knowledge, but more studies are needed before it can be used in clinical practice.

### 3.2. ECG Data and Heart Rate Signals

In another recent study, Yan and coworkers [16] explored the use of topological data analysis to classify electrocardiographic signals and detect arrhythmias. Cardiac arrhythmias are abnormal heart rhythms or irregularities in the heartbeat that may be too fast (tachycardia), too slow (bradycardia), or irregular. Arrhythmias may originate from problems with the heart’s electrical system, damage to the heart muscle, or other medical conditions. The most common types of arrhythmias are *Atrial Fibrillation* (AF) defined as an irregular and often rapid heartbeat that can lead to stroke and other heart-related complications; *Atrial Flutter* which is similar to AF, is characterized by a rapid, regular heartbeat originating in the atria; *Supraventricular Tachycardia* is characterized by episodes of rapid heart rate originating above the heart’s ventricles; *Ventricular Tachycardia*, a fast, regular beating of the heart’s lower chambers (ventricles) that can be life-threatening; *Ventricular Fibrillation* (VF) which is a chaotic, rapid heartbeat that can be life-threatening and requires immediate medical attention; and finally *Bradycardia*, a slower than normal heart rate, often caused by issues with the heart’s natural pacemaker.

In the particular case of reference [16], phase space reconstruction was used to convert the signals into point clouds, which were then analyzed using topological techniques to extract persistence landscapes as features for the classification task. The authors found that the proposed method was effective, with a normal heartbeat class recognition rate of 100% when using just 20% of the training set, and recognition rates of 97.13% for ventricular beats, 94.27% for supraventricular beats, and 94.27% for fusion beats. This ability to maintain high performance with a small training sample space makes TDA particularly suitable for personalized analysis.

One particularly difficult problem in cardiovascular disease diagnostics with important implications for therapy is the evolution of atrial fibrillation [17]. Indeed, the progression of AF from paroxysmal to persistent or permanent forms has become a significant issue in cardiovascular disorders. Information about the pattern of presentation of AF (paroxysmal, persistent, or permanent) is useful in the management of algorithms for each category, which aims to reduce symptoms and prevent severe problems associated with AF. Until now, AF classification has been based on the duration and number of episodes. In particular, changes in complexity of Heart Rate Variation (HRV) may contain clinically relevant signals of impending systemic dysregulation. HRV measures the fluctuations in the time intervals between consecutive heartbeats, providing insights into the autonomic nervous system’s activity, particularly the balance between its sympathetic and parasympathetic branches. A number of nonlinear methods based on phase space and topological properties can provide further insight into HRV abnormalities such as fibrillation. In an effort to provide a tool for the qualitative classification of AF stages, Safarbaly and Golpayegani [18] proposed two geometrical indices (fractal dimension and persistent homology) based on HRV phase space, which were able to successfully replicate the changes in AF progression.

Their studied population included 38 lone AF patients and 20 normal subjects, whose data were collected from the Physio-Bank database [19]. “Time of Life (TOL)” was proposed as a new feature based on the initial and final Čech radius in the persistent homology diagram. A neural network was implemented to demonstrate the effectiveness of both TOL and fractal dimension as classification features, resulting in a classification accuracy of 93%. The proposed indices thus provide a signal representation framework useful for understanding the dynamic changes in AF cardiac patterns but also for classifying normal and pathological rhythms.

PH was also used by Graff et al. to study HRV [20]; the authors suggested the use of persistent homology for the analysis of HRV, relying on some topological descriptors previously used in the literature and introducing new ones that are specific to HRV, later discussing their relationship to standard HRV measures. The authors showed that this novel approach produces a set of indices that may be as useful as classical parameters in distinguishing between series of beat-to-beat intervals (RR-intervals) in healthy individuals as well as in patients who have experienced a stroke.

Also in the context of fibrillation (this time for the prediction of early ventricular fibrillation), Ling and coworkers [21] proposed a novel feature based on topological data analysis to increase the accuracy of early VF prediction. In their work, the heart activity was first treated as a cardiac dynamical system, which was described through phase space reconstruction. The topological structure of the phase space was then characterized using persistent homology, and statistical features of the topological structure were extracted and defined as TDA features. To validate the prediction performance of the proposed method, 60 subjects (30 VF, 30 healthy) from three public ECG databases were used. The TDA features showed a superior accuracy of 91.7% compared to heart rate variability features and box-counting features. When all three types of features were combined as fusion features, the optimal accuracy of 95.0% was achieved. The fusion features were then ranked, and the first seven components were all from TDA features. The proposed features may have a significant effect on improving the predictive performance of early VF.

A similar approach was taken by Mjahad, et al. [22], who applied TDA to generate novel features contributing to improve both detection and classification performance of cardiac arrhythmias such as Ventricular Fibrillation (VF) and Ventricular Tachycardia (VT). The electrocardiographic (ECG) signals used for this evaluation were obtained from the standard MIT-BIH and AHA databases. The authors evaluated two types of input data for classification: TDA features and the so-called Persistence Diagram Image (PDI). When using the reduced TDA-derived features, a high average accuracy of nearly 99% was observed to discriminate between four types of rhythms (98.68% for VF; 99.05% for VT; 98.76% for normal sinus; and 99.09% for other rhythms), with specificity values higher than 97.16% in all cases. In addition, a higher accuracy of 99.51% was obtained when discriminating between shockable (VT/VF) and non-shockable rhythms (99.03% sensitivity and 99.67% specificity). These results show that the use of TDA-derived geometric features, combined with the k-Nearest Neighbor (kNN) classifier, significantly improves classification performance compared to previous works. These results were achieved without pre-selection of ECG episodes, suggesting that these features may be successfully used in Automated External Defibrillation (AED) [23,24] and Implantable Cardioverter Defibrillation (ICD) [25,26] therapies.

Jiang and collaborators studied non-invasive atrial fibrillation using TDA on ballistocardiographic (BCG) data [27]. BCG refers to the measurement and recording of the ballistic forces generated by the ejection of blood from the heart during each cardiac cycle. These forces produce subtle vibrations and movements in the body, particularly in the chest and torso. Ballistocardiography is a non-invasive technique used to capture and analyze these mechanical movements, providing valuable information about cardiac function and hemodynamics. In this research, BCG series was transformed into a high-dimensional point cloud in order to capture more rhythmic information. These point clouds were then analyzed using TDA, resulting in persistent homology barcodes. The statistics of these barcodes were extracted as nine persistent homology features to quantitatively describe the barcodes. In order to test the effectiveness of this method for detecting atrial fibrillation (AF), the researchers collected BCG data from 73 subjects with both AF and non-AF segments, and applied six machine learning classifiers. The combination of these 9 features with 17 previously proposed features resulted in a 6.17% increase in accuracy compared to using 17 features alone (p<0.001), with an overall accuracy of 94.50%. By selecting the most effective features using feature selection, the researchers were able to achieve a classification accuracy of 93.50%. According to the authors, these results suggest that the proposed features can improve AF detection performance when applied to a large amount of BCG data with diverse pathological information and individual differences.

TDA can also be combined with other data analytics/Ml approaches such as random forests (RF). Ignacio and collaborators developed a topological method to *inform* RF-based ECG classifiers [28]. In brief, in this approach, a two-level random forest model is trained to classify 12-lead ECGs using mathematically computable topological signatures as proxy for features informed by medical expertise. ECGs are treated as multivariate time series data and transformed into point cloud embeddings that capture both local and global structures and encode periodic information as attractor cycles in a high-dimensional space. Topological data analysis is then used to extract topological features from these embeddings, and these features are combined with demographic data and statistical moments of RR intervals calculated using the Pan–Tompkins algorithm for each lead to train the classifier. This multi-class classification task aims to leverage the medical expertise represented in the topological signatures to accurately classify ECGs.

The same group combined TDA and ML approaches to study atrial fibrillation in ECGs [29], showing that topological features can be used to accurately classify single-lead ECGs by applying delay embeddings to map ECGs onto high-dimensional point clouds, which convert periodic signals into algebraically computable topological signatures, thus allowing them the use of these topological features to classify ECGs.

A similar approach coupling TDA with Deep Learning to study ECGs has been recently developed [30]. Training deep learning models on time series data such as ECG poses some challenges such as lack of labeled data and class imbalance [31,32]. The authors used TDA to improve the performance of deep learning models on this type of data. The authors found that using TDA as a time-series embedding method for input to deep learning models resulted more effective than training these models directly on raw data. TDA in this context serves as a generic, low-level feature extractor that can capture common signal patterns and improve performance with limited training data. Experiments on public human physiological biosignal data sets showed that this approach leads to improved accuracy, particularly for imbalanced classes with only a few training instances compared to the full data set.

Conventional TDA of ECG time series can be further improved by considering the time delay structure to generate higher dimensional mappings of the original series [33] able to be analyzed via TDA. This was the approach taken by Fraser and coworkers [34], who found that TDA visualizations are able top unveil ectopic and other abnormal occurrences in long signals, indicating a promising direction for the study of longitudinal physiological signals.

A different approach to deal with ECG time series using TDA makes use of optimal representative cycles. In reference [35], the authors applied a topological data-analytic method to identify parts of an electrocardiogram (ECG) signal that are representative of specific topological features and proposed that these parts correspond to the P, Q, S, and T-waves in the ECG signal. They then used information about these parts of the signal, identified as P, Q, S, and T-waves, to measure PR-interval, QT-interval, ST-segment, QRS-duration, P-wave duration, and T-wave duration. The method was tested on simulated and real Lead II ECG data, demonstrating its potential use in analyzing the morphology of the signal over time and in arrhythmia detection algorithms.

### 3.3. Stenosis and Vascular Data

TDA has also been used to study stenosis, an abnormal narrowing or constriction of blood vessels. In the cardiovascular system, arterial stenosis can be particularly significant. Atherosclerosis, a condition characterized by the buildup of plaque on the inner walls of arteries, is a common cause of arterial stenosis. Nicponski and collaborators [36] demonstrated the use of persistent homology to assess the severity of stenosis in different types of stenotic vessels. They introduced the concept of *critical failure value*, which applies one-dimensional homology to these vessels as a way to quantify the degree of stenosis. They also presented the spherical projection method, which could potentially be used to classify various types and levels of stenosis, and showed that the two-dimensional homology of the spherical projection can serve as a new index for characterizing blood vessels. It is worth noticing that, as in many other instances in data analytics, data pre-processing often represents a crucial stage [37].

### 3.4. TDA in Echocardiography

Applications of TDA to echocardiographic data have also been developed. Such is the case of the work the group of Tokodi [38,39] who analyzed a cohort of 1334 patients to identify similarities among patients based on several echocardiographic measures of left ventricular function. Echocardiography, a medical imaging technique that uses sound waves to create detailed images of the heart, generates 2D and 3D maps of vascular structure. However, accurate image reconstruction is far from trivial, in particular for small convoluted cavities. A network was developed in reference [38] to represent these similarities, and a group classifier was used to predict the location of 96 patients with two consecutive echocardiograms in this network. The analysis revealed four distinct regions in the network, each with significant differences in the rate of major adverse cardiovascular event (MACE) rehospitalization. Patients in the fourth region had more than two times the risk of MACE rehospitalization compared to those in the other regions. Improvement or stability in Regions I and II was associated with lower MACE rehospitalization rates compared to worsening or stability Regions III and IV. The authors concluded that TDA-driven patient similarity analysis may improve the precision of phenotyping and aid in the prognosis of patients by tracking changes in cardiac function over time.

## 4. Conclusions

We showed that within the realm of cardiovascular signal analysis, the utilization of Topological Data Analysis (TDA) displayed remarkable promise across diverse applications (see Table 2). As we mentioned, Aljanobi and Lee applied the Mapper algorithm to predict heart disease with exceptional accuracy, achieving 99.32% accuracy in the Cleveland data set and 99.62% in the Statlog data set. This underscores the effectiveness of TDA in identifying significant features for disease prediction. Moving beyond disease prediction, TDA proved invaluable in the risk stratification of patients post-acute myocardial infarction (AMI). Lopez and colleagues employed the Mapper algorithm on Electronic Health Records (EHR), successfully classifying distinct subsets of AMI patients. The insights gained into high-risk subgroups and their susceptibility to adverse events up to 3 years post AMI showcase the potential of TDA in personalized patient care.

The application of TDA to electrocardiographic (ECG) signal classification yielded notable outcomes. Yan and Ling explored TDA for arrhythmia detection and early ventricular fibrillation prediction, respectively. Yan’s use of persistent homology achieved high recognition rates, while Ling’s approach demonstrated superior accuracy compared to traditional features, indicating the robustness of TDA in diverse ECG applications. Safarbaly and Golpayegani delved into the progression of atrial fibrillation (AF) using persistent homology. Their novel geometric indices, based on HRV phase space, successfully replicated changes in AF stages, attaining a classification accuracy of 93%. This not only sheds light on AF dynamics, but also presents a potential tool for qualitative AF classification.

The synergy of TDA with machine learning was evident in Mjahad’s work, where TDA-derived geometric features significantly improved the classification of cardiac arrhythmias. The application extended to non-invasive atrial fibrillation detection by Jiang and collaborators, showcasing an enhanced accuracy compared to traditional features alone. Byers and team explored the integration of TDA with deep learning for ECG classification. Using TDA as a time-series embedding method for deep learning models resulted in improved accuracy, particularly for imbalanced classes, addressing challenges in classification tasks with limited training data. TDA’s versatility extended to stenosis severity assessment in various vessels, as demonstrated by Nicponski and collaborators. The critical failure value and spherical projection methods provided quantifiable measures, highlighting the potential of TDA in characterizing blood vessels. Lastly, TDA was applied to echocardiographic data by Tokodi and team for patient similarity analysis. The Mapper algorithm revealed distinct regions in a network representing similarities among patients based on echocardiographic measures. The identified regions were associated with significant differences in major adverse cardiovascular events (MACE) rehospitalization rates, offering insights into patient prognosis.

In conclusion, the diverse applications of TDA in cardiovascular signal analysis underscore its effectiveness in disease prediction, risk stratification, signal classification, and patient similarity analysis. The integration of TDA with machine learning and deep learning techniques presents exciting avenues for advancing our understanding of complex cardiovascular patterns and improving clinical outcomes (for software implementations see Table A1). However, further research and validation are essential to solidify the clinical applicability of these TDA-based approaches.

### Perspectives and Limitations

We presented a panoramic (and by necessity incomplete) view of the applications of topological data analysis to the study of cardiovascular signals. We must, however, stress that, as every analytic technique, TDA has a set of limitations and a range of applicability. Among its limitations, we can include its inherent complexity both to implement the analyses and to interpret the results. Also relevant is TDA’s sensitivity to noise and reliance on assumptions of the data distribution (for instance, one assumes that filtration can be carried out unambiguously). Another aspect to consider is that topological data analysis can be computationally intensive, particularly for large or complex data sets. This, along potential biases in noisy signals, may preclude the accurate identification of the topological features of certain data sets, especially if the data are of low quality. As we said, topological data analysis relies on assumptions about the data set, for instance, the existence of a *natural* distance measure. This condition may not hold in all cases. This may potentially limit its applicability. Additionally, the results of topological data analysis can be difficult to interpret, particularly for non-experts, a fact that can make challenging to communicate the results of a topological analysis to a general audience. However, even in light of these limitations, TDA is a quite powerful approach that (perhaps in combination with other data analytic approaches) may result very useful for the study of complex biosignals, such as those arising in cardiology. 

## Figures and Tables

**Figure 1 entropy-26-00067-f001:**
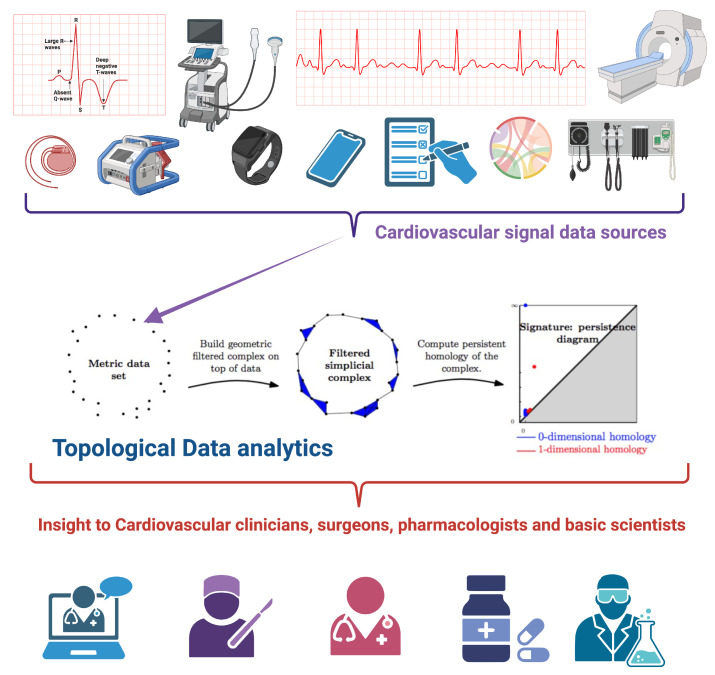
Topological Data Analysis in cardiology. Starting from diverse sources of cardiovascular signals, topological data analytics allowed a systematic study and categorization leading to a better understanding of the underlying phenomena, thus providing clues for clinicians and basic scientists. Figure generated with BioRender.com, central panel is taken from Larrysong, CC BY-SA 4.0 https://creativecommons.org/licenses/by-sa/4.0 (accessed on 4 June 2023), via Wikimedia Commons.

**Figure 2 entropy-26-00067-f002:**
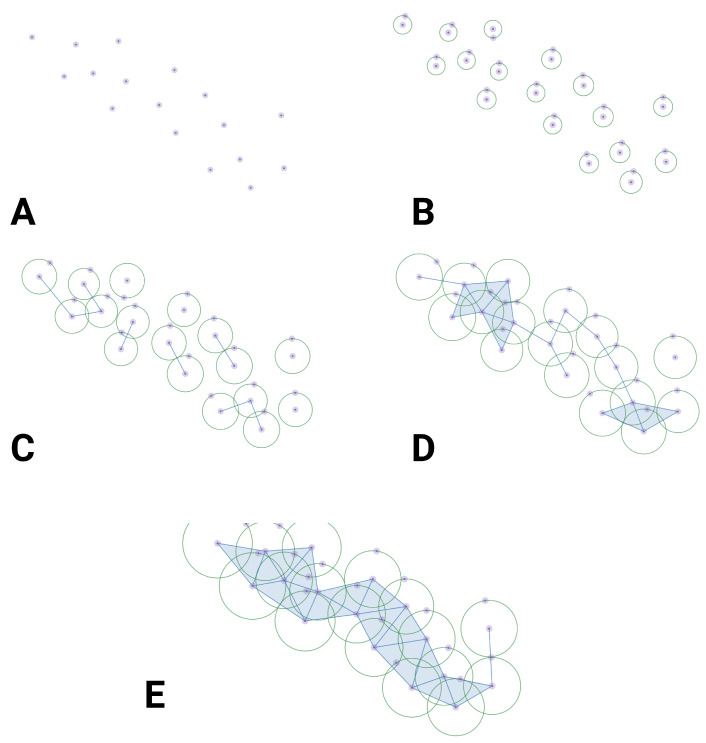
**Persistent homology via FSC**. Panel (**A**) presents a set of data points in feature space. In panel (**B**), we put a ball of radius ϵ around each data point. In panels (**B**–**E**), we increase radius ϵ. Neighborhoods start to overlap, giving rise to the establishment of a filtered simplicial complex.

**Figure 3 entropy-26-00067-f003:**
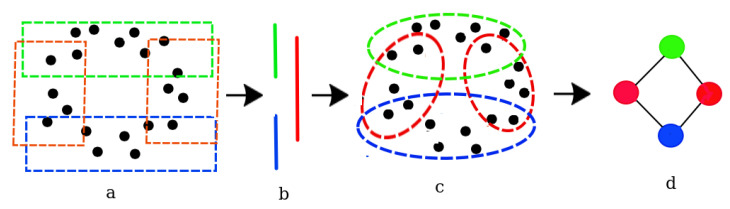
The steps of the Mapper algorithm. (**a**) The data, a cloud of points. (**b**) The projection of the data into a lower dimension space. (**c**) The preimage is clustered and (**d**) a graph is built based on the clustered groups. See text.

**Figure 4 entropy-26-00067-f004:**
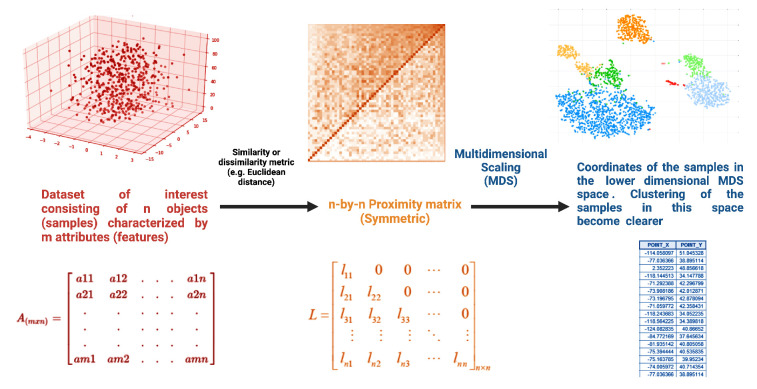
Classic multidimensional scaling.

**Table 1 entropy-26-00067-t001:** Some types of covering maps and their main applications.

Covering Map	Type of Data	Brief Description
Vietoris–Rips Complex	Point cloud data, particularly when dealing with metric spaces. It is often used in applications like sensor networks, molecular chemistry, and computer graphics	The Vietoris–Rips complex connects points in the data if they are within a certain distance (the radius parameter) of each other, forming simplices (e.g., edges, triangles, tetrahedra) based on pairwise distances
Čech Complex	Similar to the Vietoris–Rips complex, Čech complexes are used for point cloud data in metric spaces	The Čech complex connects points if they belong to the same open ball of a specified radius. It can capture similar topological features as the Vietoris–Rips complex but may have a different geometric structure
Alpha Complex	Alpha complexes are useful for point cloud data in metric spaces and provide an alternative representation of the topological structure	The alpha complex connects points with a Delaunay triangulation, considering balls whose radii can vary at each point to ensure that the complex is a subcomplex of the Vietoris–Rips complex
Witness Complex	Witness complexes are used for point cloud data but are particularly useful when dealing with data that may not be uniformly sampled or when dealing with non-metric or qualitative data	Witness complexes are constructed by selecting a subset of witness points from the data. Each witness point witnesses the presence of other data points within a specified distance. This can be used to capture topological features in a more robust way, especially when data is irregular
Mapper	Mapper is a flexible approach that can be applied to various types of data, including both metric and non-metric spaces	Mapper is not a traditional covering map but rather a method for creating a topological summary of data by combining clustering and graph theory. It can be adapted to different data types and is useful for exploratory data analysis
Rips Filtration and Čech Filtration	These are extensions of the Vietoris–Rips and Čech complexes, respectively, that allow for the analysis of topological features at different scales	By varying the radius parameter continuously, Rips filtration and Čech filtration produce a sequence of simplicial complexes. This can be useful for capturing topological features at different levels of detail and studying persistence diagrams

**Table 2 entropy-26-00067-t002:** Comparison of TDA Models in Cardiovascular Signal Applications.

Study	Model Used	Application	Data Set	Features/Methods	Accuracy/Performance
Aljanobi and Lee [5]	Mapper Algorithm	Heart Disease Prediction	UCI Heart Disease Data sets (Cleveland, Statlog)	Nine significant features, tri-dimensional SVD filter	Cleveland: 99.32%, Statlog: 99.62%
Lopez et al. [15]	Mapper Algorithm	Risk Stratification of AMI Patients	EHR of 798 AMI subjects	Clinical variables from EHR, Mapper algorithm	3-year AE rates stratification
Yan et al. [16]	Persistent Homology	Arrhythmia Detection	ECG Signals	Phase space reconstruction, persistence landscapes	Normal: 100%, Ventricular: 97.13%, Supraventricular: 94.27%
Safarbaly and Golpayegani [18]	Persistent Homology	AF Progression	Physio-Bank Database	Fractal dimension, persistent homology	Classification accuracy: 93%
Graff et al. [20]	Persistent Homology	HRV Analysis	Healthy individuals and post-stroke patients	Topological descriptors, comparison with standard HRV measures	Distinction between RR-intervals
Ling et al. [21]	Persistent Homology	Early Ventricular Fibrillation Prediction	Public ECG Databases	Phase space reconstruction, persistent homology, statistical features	Accuracy: 95.0%
Mjahad et al. [22]	Persistent Homology	Arrhythmia Classification	MIT-BIH and AHA Databases	TDA features and Persistence Diagram Image (PDI)	Accuracy: 99%, Sensitivity: 99.03%, Specificity: 99.67%
Jiang et al. [27]	Persistent Homology	Non-invasive AF Detection	BCG Data from 73 subjects with AF and non-AF segments	TDA on BCG data, machine learning classifiers	Classification accuracy: 94.50%
Ignacio et al. [28]	TDA (Informed Random Forests)	ECG Classification	12-lead ECGs	Topological signatures, random forests	Not specified
Ignacio et al. [29]	TDA (Informed Random Forests)	Atrial Fibrillation Classification	Single-lead ECGs	Delay embeddings, topological features	Not specified
Byers et al. [30]	TDA combined with Deep Learning	ECG Classification	Public human physiological biosignal data sets	TDA as time-series embedding for deep learning models	Improved accuracy for imbalanced classes
Fraser et al. [34]	TDA with Time Delay Structure	ECG Visualization	Longitudinal physiological signals	Time delay structure, TDA visualizations	Unveiling abnormal occurrences
Dlugas et al. [35]	TDA with Optimal Representative Cycles	ECG Signal Morphology Analysis	Simulated and real Lead II ECG data	Identification of P, Q, S, T-waves, measurement of intervals	Not specified
Nicponski et al. [36]	Persistent Homology	Stenosis Severity Assessment	Various types of stenotic vessels	Critical failure value, spherical projection, 2D homology	Quantification of stenosis severity
Tokodi et al. [38]	Mapper Algorithm	Patient Similarity Analysis	Echocardiographic measures of left ventricular function	Network representation, group classifier	Prognosis of patients

## Data Availability

Not applicable.

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
