# Peer review of "Topological Data Analysis in Cardiovascular Signals: An Overview"

_entropy, 2024, doi:10.3390/e26010067_

Round 1

Reviewer 1 Report (Previous Reviewer 2)

Comments and Suggestions for Authors

The authors did a good job of improving the paper.

Author Response

The authors did a good job of improving the paper.

The authors are grateful to Reviewer 1 for the assessment of our revised manuscript.

Reviewer 2 Report (New Reviewer)

Comments and Suggestions for Authors

The present manuscript presents an overview about the Topological Data Analysis of Cardiovascular diseases. Major part of the paper concentrate on the definitions of TDA and the different uses of several strategies. This could be useful for researchers in this field. 

The application to the CV problem, however, is more descriptive and it is difficult to get any usefull information. Most of the results in section 3 could be presented in form of a table to compare the accuracy for the distint ML models - KNN, RF, Deep Learning,... 

It could be also summarized in a new section 4: Conclusions.

Comments on the Quality of English Language

I have not find any significative english misspelling, anyway you could revise it overall.

Author Response

The present manuscript presents an overview about the Topological Data Analysis of Cardiovascular diseases. Major part of the paper concentrates on the definitions of TDA and the different uses of several strategies. This could be useful for researchers in this field.

The authors want to thank Reviewer 2 for their scholarly analysis of our manuscript. 

The application to the CV problem, however, is more descriptive and it is difficult to get any useful information. Most of the results in section 3 could be presented in the form of a table to compare the accuracy for the distinct ML models - KNN, RF, Deep Learning,... 

It could be also summarized in a new section 4: Conclusions.

Thank you for your suggestion. We have added both, a Table at the end of section 3 and a new section 4 containing some concluding remarks as you suggested.

Reviewer 3 Report (New Reviewer)

Comments and Suggestions for Authors

1. The abbreviation of FSH is lost.

2. Should the MDS techniques including in the TDA technique? Since in the previous literatures usually the included tools are PH and Mapper, the author should explain their considerations.

3.From my viewpoint, this review entitled with TDA in cardiovascular signals, however, the author included the EHR-related contents adopted Mapper. The text-based information might not be 'signal' thing, so the author should reconsider it.

4. Meanwhile, the cardiovascular application part should include some of the basic ideas about the data, not just stacking some literatures used some TDA technique. The author should include some introductions of the topic on the cardiovascular data/signals themselves.

5.Nearly 2/3 of your work is focus on TDA itself, the application of TDA seems to short considering the works focuses on the cardiovascular use of TDA. My suggestion is that considering developing the application descriptions from the data type site, and give some basic clues of how the technique was used in such cardiovascular data. The current application descriptions are not clear enough to the readers, which should reorganized carefully. 

Comments on the Quality of English Language

Should be fine to me.

Author Response

The authors are grateful to Reviewer 3 for their thorough reviewing of our work. We will now present a point-by-point respond to your comments and requests.

1. The abbreviation of FSH is lost.
Fixed it. It was a typo for FSC, filtered simplicial complex.

2. Should the MDS techniques be included in the TDA technique? Since in the previous literature usually the included tools are PH and Mapper, the author should explain their considerations.

This section was suggested in a previous stage of reviewing. Though not exactly a “topology-based” method, we believe that it indeed introduces (in particular the associated figure) some ideas, such as data transformation and dimensionality reduction by feature selection, in a simpler way for non-specialist readers. It is also commonly applied along TDA methods. We have added a brief clarification in the revised version of the manuscript.

3.From my viewpoint, this review is entitled with TDA in cardiovascular signals, however, the author included the EHR-related contents adopted Mapper. The text-based information might not be 'signal' thing, so the author should reconsider it.

Most of the content is actually related to signals. After some pre-processing (that we passed-by to include) one can also apply TDA techniques to structured data (with some limitations). To avoid confusion we have added a brief clarification in the corresponding section of the manuscript.

4. Meanwhile, the cardiovascular application part should include some of the basic ideas about the data, not just stacking some literature using some TDA technique. The author should include some introductions of the topic on the cardiovascular data/signals themselves.

Thank you for your suggestion. We have added brief introductory paragraphs whenever needed.

5.Nearly 2/3 of your work is focus on TDA itself, the application of TDA seems to short considering the works focuses on the cardiovascular use of TDA. My suggestion is that considering developing the application descriptions from the data type site, and give some basic clues of how the technique was used in such cardiovascular data. The current application descriptions are not clear enough to the readers, which should reorganized carefully. 

This issue is complementary with the previous one. To comply with your suggestions we have significantly enhanced/rewritten some sections of the manuscript.

Round 2

Reviewer 3 Report (New Reviewer)

Comments and Suggestions for Authors

1. All my concerns are addressed. One minor suggestion is that ref [16] should modified into its peer-reviewed version as shown in https://ieeexplore.ieee.org/document/10331360

This manuscript is a resubmission of an earlier submission. The following is a list of the peer review reports and author responses from that submission.

Round 1

Reviewer 1 Report

Comments and Suggestions for Authors

The paper is a review paper on TDA methods and not offering any new information. There are a lot of TDA review papers in literature and this paper is not offering anything new.

Comments on the Quality of English Language

English is OK.

Reviewer 2 Report

Comments and Suggestions for Authors

The review introduces the main principles of topological data analysis (TDA) methods and discusses three main algorithms: persistent homology, the Mapper algorithm, and multidimensional scaling. In addition, the authors present applications of TDA methods for analyzing cardiovascular signals, such as electrocardiogram, and assessing pathologies like atrial fibrillation, ventricular fibrillation, ventricular tachycardia, and others, based on papers by other researchers. The authors summarize that TDA methods hold great potential in the diagnosis of cardiovascular diseases, but they also acknowledge that there are challenges, such as computational intensity and the difficulty of interpreting results. However, the review could benefit from providing more specific information to help readers better understand the principles of the described methods.

1.      Based on the title and content of the study, the manuscript should be assigned to the Review section, rather than the Article section.

2.      The Keywords section should include additional keywords, such as persistent homology and the Mapper algorithm, which are not provided in the title.

3.      Although the abstract mentions photoplethysmography and arterial stiffness, the manuscript only discusses electrocardiography and ballistography. This discrepancy should be addressed.

4.      The introduction contains repetitive information about unsupervised, supervised, and reinforcement learning algorithms. The emphasis should be shifted towards providing a more comprehensive overview of TDA methods.

5.      More illustrations should be included to help non-experts understand TDA techniques. Currently, there are only two illustrations in the manuscript.

6.      The manuscript should provide a clear discussion of the advantages and disadvantages of the Vietoris-Rips strategy, the witness strategy, and the lazy-witness strategy methods.

7.      The manuscript should provide a more detailed description of how the persistent homology group is calculated, including the parameters that are extracted and how they should be interpreted (section 2.1.2).

8.      The manuscript should provide an explanation of how to interpret topological features such as "connectedness", "loops", "holes", and "voids".

9.      Additional illustrations should be included to explain the Mapper and multidimensional scaling methods in greater detail.

10.   The manuscript should discuss the characteristics of data sets that are most relevant when choosing a covering map and identify the most important factors to consider.

11.   The manuscript should provide guidance on when it is most appropriate to use classical, nonmetric, and metric multidimensional scaling methods.

Comments on the Quality of English Language

Moderate style improvement is needed.